# Dentigerous Equine Teratoma in a Stallion: Surgical Management and Clinicopathology

**DOI:** 10.3390/vetsci8050084

**Published:** 2021-05-13

**Authors:** Leonardo Leonardi, Alice Bertoletti, Enrico Bellezza, Ilaria Pettinari, Rodolfo Gialletti

**Affiliations:** Department of Veterinary Medicine, University of Perugia, Via San Costanzo, 4—06126 Perugia, Italy; alicebertoletti1@gmail.com (A.B.); enrico.bellezza@unipg.it (E.B.); pettinariilaria@hotmail.it (I.P.); rodolfo.gialletti@unipg.it (R.G.)

**Keywords:** teratoma, testicle, teeth, equine, cryptorchidism, cancer, laparoscopy

## Abstract

Teratoma is a rare germ cell testicular cancer composed of cells that are not normally present in the site where it originates. These tumors are rarely described in horses, where they may develop due to cryptorchidism. Teratomas consist of cells originating from different germinal layers, arising from germinal multi-potential cells with differentiation defects, and can simultaneously contain several tissues from two or more embryonic layers. Testicular teratomas are described in horses, cats, dogs, wild boars, bulls, and humans. In the rare descriptions found in literature of testicular teratoma in stallions, they occur frequently in cryptorchid testicles, as a consequence of congenital neoplasm. To our knowledge there is no other report of a dentigerous equine teratoma in a stallion. We describe here a successful laparoscopic removal of a testicular teratoma and its clinic-pathological features.

## 1. Introduction

The word teratoma was coined by Rudolph Virchow in 1863 from the Greek words “teras and onkoma” that respectively mean “monster and tumor [swelling]”. Teratoma is a rare embryonic tumor that can be both benign or malignant, most often diagnosed as benign. Teratoma is composed of mature somatic tissue that is not well organized and composed of neoplastic germ cells [1] originating from all three layers of embryonic germ cells: ectoderm, mesoderm and endoderm (ex: epithelium, brain, bone, cartilage) [1,2,3,4,5,6,7,8]. In human medicine these tumors that originate from all three germ layers are also called “tridermomas”; this term is not currently used in veterinary medicine. Furthermore, it has been demonstrated that most of these neoplasms derive from anomalies in the ectodermal layer and in a minority from the endodermal layer. Teratomas can be classified as mature and immature based on cell differentiation, therefore they can consist of different mature and immature tissues derived from the three layers of germ cells [9]. Testicular cancer accounts for only 1% of malignant tumors in both humans and animals [10]. Testicular teratomas are rare in horses, and are generally diagnosed accidentally during autopsy or surgery for cryptorchidism, an incomplete descent of one or two testicles in the scrotum. As they do not have any clinical manifestations, it is usually an incidental finding. A greater probability of detecting teratomas in horses occurs when testicles are localized in the abdomen. Cryptorchid testicles commonly remain localized in the abdominal cavity, where they are more prone to the onset of neoplastic conditions such as Seminomas, Sertoli cell tumors and Leydig cell tumors; these neoplasms have an approximate equal frequency in mammalian species [11,12,13,14].

To date, there is no effective therapy for this tumor, other than surgical removal. In this report we describe the clinical and histopathological characteristics of a rare case of unilateral abdominal testicular dentigerous teratoma in a horse.

## 2. Case Report

A five-year-old, 450 kg, thoroughbred stallion was admitted for standing laparoscopy for cryptorchidectomy. The left testis was in the scrotum but the right testis was not palpable. The horse was for 48 h and before surgery, sodium penicillin (22,000 IU/kg IM) and flunixin meglumine (1.1 mg/kg IV) were administered as per our pre-operative protocols. The horse was restrained and the right flank was clipped and aseptically prepared for surgery. Sedation was achieved with a continuous rate of infusion of detomidine hydrochloride (8.4 μg/kg IV) and butorphanol tartrate (10 μg/kg IV). The first portal was positioned midway between the tuber coxae and the 18th rib, at the dorsal border of the internal abdominal oblique muscle. The second portal was placed 5 cm ventral and 3 cm caudal to the first one, and the third portal was located in the 17th intercostal space at the level of the ventral aspect of the tuber coxae. A 15-mm skin incision was made at each site for laparoscope and instrument portals, following the infiltration of 20 mL of 2% mepivacaine subcutaneously. A 12-mm trocar-cannula unit (VersaOne^TM^, Medtronic-Minneapolis, MN, USA) was introduced at portal one into the abdomen with a 10-mm laparoscope 0° inside to provide a safe access. The abdomen was then distended using an automatic high-flow CO_2_ insufflator up to an intra-abdominal pressure of 8–10 mmHg. The accessory trocars were inserted at portals two and three under laparoscopic supervision. The laparoscope was then placed in the third portal and directed caudally toward the vaginal ring. Only a portion of the mesorchium with the spermatic cord and the ducts deferens was visualized, since the identification of the retained testis was not possible due to the interposition of the large colon. The mesorchium was desensitized with 20 mL of 2% mepivicaine via a laparoscopic needle inserted at the first portal, before starting its manipulation. Ten-millimeter Teeth Claw Jaw Grasping Forceps were inserted through portal two, and placed at the more distal portion of the spermatic cord, which was held under slight tension. A 10-mm Ligasure^TM^ was inserted through portal 1 and was applied to seal and transect the spermatic cord and ducts deferens just dorsal to the grasping forceps. The mesorchium, freed of its dorsal attachment, was pulled up, revealing the presence of a large mass, compatible with the right retained testis. The Ligasure^TM^ was removed and another of the 10-mm Teeth Claw Jaw Grasping Forceps was placed at portal one; this grasped the spermatic cord as close as possible to the testis, which was then removed from the abdomen by enlarging the incision up to 10 cm. The abdominal cavity was checked for any abnormal bleeding, then it was deflated and the trocars were removed. Portal one was closed in two layers: the external abdominal oblique muscle was closed with size 0 polydioxanone suture and the skin with size 0 polypropylene. Only the skin of portals two and three were closed with size 0 polypropylene in a simple interrupted pattern. The retained testis was submitted for histopathology examination, while the other was left in the scrotal cavity by the will of the owner. Post-operatively, the horse received sodium penicillin (22,000 IU/kg IM every 12 h) and flunixin meglumine (1.1 mg/kg IV every 24 h) for 5 days as required by our post-operative prevention and prophylaxis protocols, to avoid complications of a mainly infectious nature. The horse was confined then to a stable for one week with daily hand walking. He was then discharged and normal activity resumed.

### 2.1. Macroscopic Features and Methods

Once the testicle was extracted from the abdomen, considering its anomalous shape and its hard and non-deformable state of consistency, we decided to take radiographs. The teratoma appeared to predominantly involve the structure of the epididymis. Macroscopic examination showed the removed testicle with an enlarged oval shape of 9 × 6.5 × 6 cm, with a very hard “bony” compact consistency due to the internally formed component (teeth). After a necessary cross cut made with a band saw, the cut surface allowed us to detect the presence of four morphologically distinct elements, whitish in color, hard, and apparently normal shaped, referable to well differentiated dental structures, immersed in a hard dentinal-osseous amorphous matrix, which occupied most of the entire cut surface, replacing what must have been the normal structural tissue of the testis (Figure 1 and Figure 2).

Several tissue samples were collected and fixed in 10% buffered formalin, decalcified with aqueous solution of formic and hydrochloric acid for more than 20 days; subsequently, they were embedded in paraffin, cut into sections of 4–5 µm, and then stained with Hematoxylin-Eosin.

### 2.2. Histological Features

In general, the histopathological investigations revealed a neoplastic lesion composed mainly of different types of mature tissues by many germinal layers. In this case bone, enamel, dentin and mineralized areas represented the predominant part of the tumor which were also associated with separate and more marginal foci of mature fibrous tissue and tubular-like and glandular-like structures lined with mucosa, arranged in a markedly disorganized way, unlike the teeth which showed a well-ordered structural organization especially in the enamel and dentin components. In this case, no malignancy was detected at histopathological investigations [Figure 3].

## 3. Discussion

Reproductive disorders are common in horses, and cryptorchidism is one of the most common conditions where surgical approaches to remove the testes are the most effective and low risk treatment.

Teratomas are tumors consisting of mature and immature tissues derived from pluripotent cells of all three germ layers, and which arise from multipotential germ cells that have undergone partial differentiation. In teratomas, ectodermal (skin derivatives and neural tissue), mesodermal (bone, fat, cartilage and muscle) and endodermal (gastrointestinal and bronchial epithelium, thyroid) tissues can coexist.

Teratoma occurs mostly in cryptorchid testicles in horses, where it is reported in an age range of under one year to a maximum of five years, considering it is most likely congenital [15,16,17,18]. It can be macroscopically differentially diagnosed with tumors such as extraskeletal osteosarcoma, adamantinoma, and undifferentiated teratocarcinoma, but histopathology analysis is able to clarify the exact nature of the tumor under examination [19].

## 4. Conclusions

Based on our information, this case-report represents the first case described in literature of a dentigerous equine teratoma in this species. Our bibliographic research found only very few cases of teratomas with ectopic teeth described in the human species, mainly localized in the orbital region or in the lung, and never at testicular level and with related primitive onset. Laparoscopy is the best surgical technique to approach this type of condition because it is minimally invasive and offers good visibility with very few and infrequent complications as described by D. Hendrickson [20].

To our knowledge the surgery was well tolerated by the horse [20], which is in good health and has returned to regular racing performance. The other scrotal testicle was left at the behest of the owner and showed no clinical signs of disease.

## Figures and Tables

**Figure 1 vetsci-08-00084-f001:**
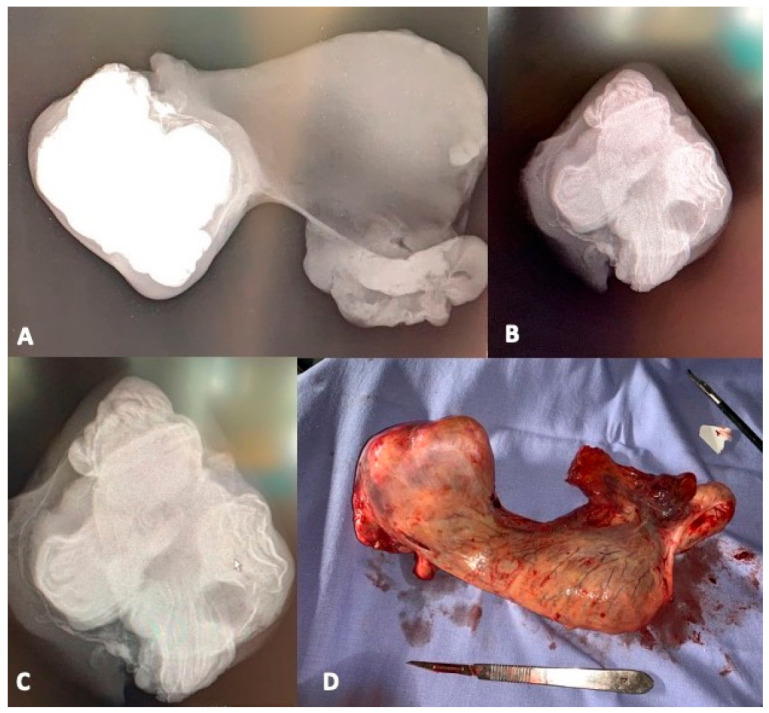
Radiographic examination of the excised testicle removed from the abdomen. Presence of an irregular radiopaque area on the left of the mass and soft-tissue opacity on the right (**A**). Particular of the left area seen in figure A showing the presence of multiple teeth-like calcifications (**B**,**C**). Anatomical appearance of the retained testicle (**D**).

**Figure 2 vetsci-08-00084-f002:**
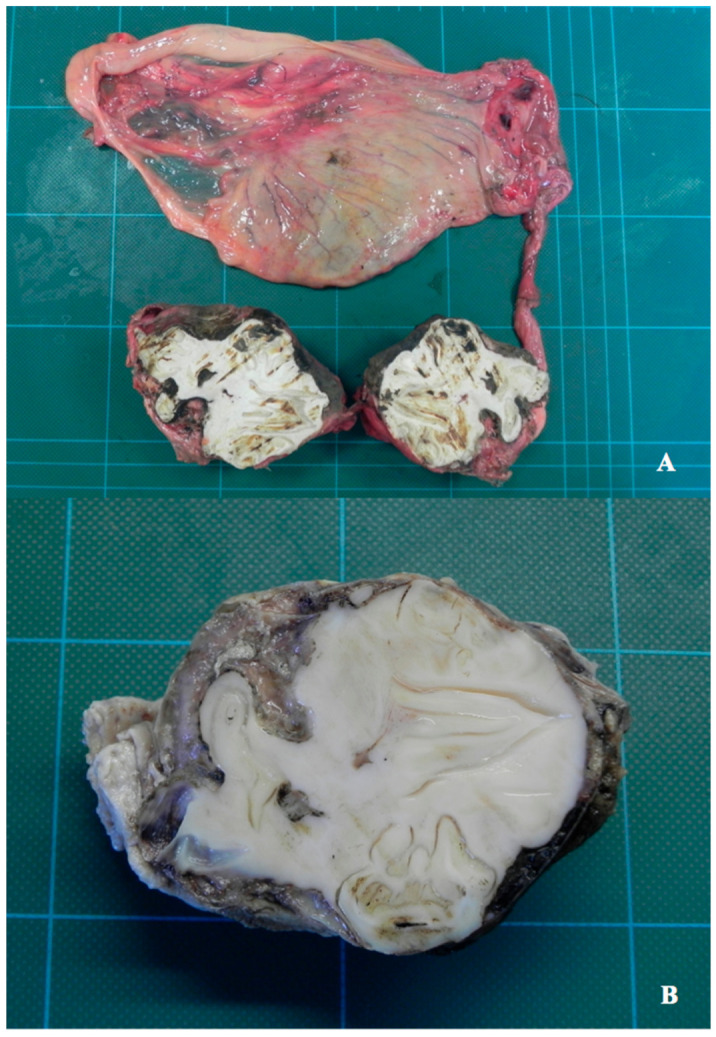
Gross picture of the tumor. Cut surface after resection (**A**,**B** [higher magnification]) and after careful cleaning of the surface, previously altered by cutting operations with an electric saw due to the extreme hardness of the tumor.

**Figure 3 vetsci-08-00084-f003:**
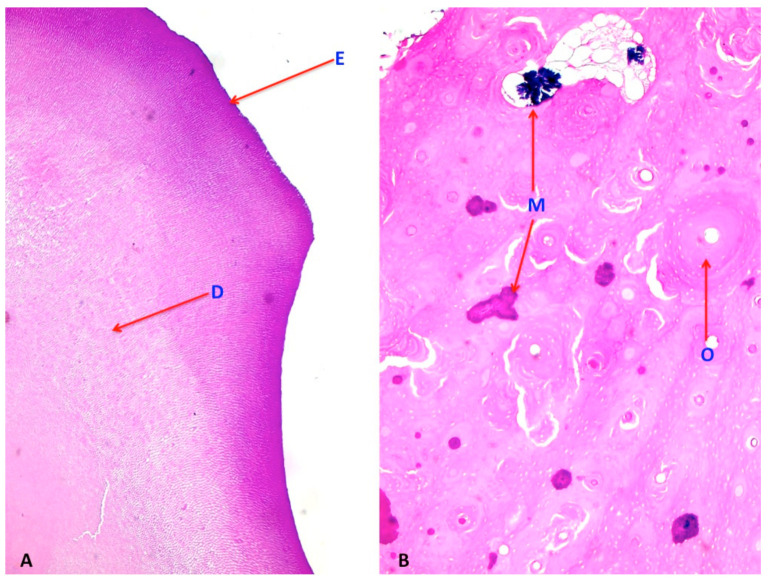
(**A**) Histological picture stained with hematoxylin-eosin of a part of a dental structure isolated from the testicle, showing the presence of enamel [E] and dentin [D], the latter with the typical structure of oriented parallel bundles perpendicularly. (**B**) Histological picture of the testicular tumor where it is possible to detect a periodontal bone component, adhering to the root structure of the testicular tooth, and constituting the alveolar bone at different stages of maturation and with a compact appearance, with well differentiated osteonal structures [O] and areas inside the bone tissue at different stages of maturation and mineralization [M].

## Data Availability

The data presented in this study are available in the manuscript.

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
