# Peer review of "Dentigerous Equine Teratoma in a Stallion: Surgical Management and Clinicopathology"

_vetsci, 2021, doi:10.3390/vetsci8050084_

Round 1

Reviewer 1 Report

The paper reports an unpublished testicular teratoma with teeth what is interesting. But it is necessary to better it. The reading is harder and the paper is badly organized. First of all is necessary to edit the English by a native speaker.

There are many references of the human medicine literature but I miss references about temporal teratoma (dentigerous cyst, frequent in horses), others about testicular teratomas, and ovarian teratomas, both in horses. Please, comment that there are ovarian teratomas with teeth. Also, you should include any laparoscopic cryptorchidectomy reference.

2.1 Macroscopic features and methods: You put radiographic images but is not commented that you took them. It´s important to describe properly the macroscopic characteristics of the testis together the teratoma. Where exactly the teratoma was localized? In the testis, epididimis, etc.

Do you have any laparoscopic image of the teratoma?

The discussion should be extended. 

Below I note some corrections in the different lines.

line 37 as like as also to diagnosed its .....as like as to diagnose it

line 38 because they don’t cause often clinical signs and they are.....

because it doesn’t cause often clinical signs and it is 

line 55 butorphanol, is correct the dose (10 microgr/k)? I think is lower

line 90 macroscopic.......Macroscopic

line 92  component (teeth) (Fig.). After   ........Remove (Fig.)

line 95 immersed in a with hard............I think with should be deleted

line 107  After a long period of treatment of the samples to softening and decalcifying the tissues, it was possible to obtain....

Decalcifying was previously commented.   Delete all sentence

line 135 we have been consider..........considered

Reference 20 is an inappropriate self-citation

Author Response

First of all we thank the Reviewers for their precious and generous comments on our manuscript.

We have edited it to address all their concerns.

We believe and hope that now manuscript is sutitable fot publication in Journal of Veterinary Sciences.

REVIEWER 1:

• We edit the english.

• Referring to other references of the human medicine: we have repeatedly searched for new

bibliographic data relating to temporal teratomas, dentigerous cysts, etc. but we found no significant

data in the horse. We have found this work "Bilateral dentigerous cysts (heterotopic polyodontia) in a

yearling Standardbred colt", L. C. R. Smith, S. T. Zedler, S. Gestier. S. E. Keane, W. Goodwin, A. W. van

Eps, Equine veterinary education, 2011, 573-578, that does not seem very relevant to the work, but if

the Reviewer deems it necessary we can also insert it in the text and in the literature. Referring to

Cryptorchidectomy reference we can enclose also this paper, if the Reviewer think necessary:

"Laparoscopic cryptorchidectomy and ovariectomy in horses", D. Hendrickson, Veterinary clinics

equine practice, 22, 2006, 777-798., but this is generic description of normal procedures.

• once the testicle was extracted from the abdomen, considering its anomalous shape, its hard and

non-deformable state of consistency, we preferred to make radiographs, which appeared to us as the

ideal method for a better description of even the details. The teratoma appeared to predominantly

involve the structure of the epididymis, as we have also added in the text.

• we do not have laparoscopic images because at the first laparoscopic examination it was possible to

visualize only the apparently normal part of the testicle. The testicle was then isolated with ligasure

and only after it was completely removed and extracted did we realize that it looked abnormal.

• We add that: laparoscopy is the best surgical technique for approach and resolution of this type of

injury because it is minimally invasive and with very few and infrequent secondary complications as

like as described by D. Hendrickson.

• Line 55 Butorphanol: it is our practice to use these doses on a regular basis. In horses.

• We did all corrections in the different lines.

• We delete reference 20

We remain at your complete disposal for any further need.

Many thanks

Leonardo Leonardi et al.

Reviewer 2 Report

Please find attached file for comments

Author Response

I would like to thanks for the suggestions and corrections sent by your Reviewers.   We have tried to make all the suggested changes, also expanding the discussion part. About the corrections sent by the reviewer 2, we didn’t use the term OLD (stallion) as she/he suggested in specific comments, lines 15-17, because the Horse was 5 yo and other Reviewers rightly ask to delete this word.

Reviewer 3 Report

All comments and remarks on pdf file.

Overall, manuscript needs English proof editing. Extensive text corrections and rephrasing are mandatory.

About the case itself, there is no description of the clinical status of the horse, clinical signs? owner complain? activity? And of course, a 5 year old horse is NOT and old stallion, but rather a young animal.

Are there any images during the laparoscopy? Additionally, nothing is mentioned about the other testis? was it removed? analyze ? yes no why?

Laparoscopy infection rates are reported to be close to 0%. Why was the horse keep on antibiotic therapy for five days? were there signs of infection, fever? subcutaneous emphysema?

Author Response

First of all we thank the Reviewers for their precious and generous comments on our manuscript.

We have edited it to address all their concerns.

We believe and hope that now manuscript is sutitable fot publication in Journal of Veterinary Sciences.

REVIEWER 2:

• We edit the english.

• Referring to clinical status and data of the horse: The horse was referred to the clinic due to the need

for removal of the right testicle retained in the abdomen. The other testicle appeared regularly

contained within the scrotum, as indicated. The owner expressed his firm will to leave the normally

descended tissue in the scrotum and to surgically remove only the testicle retained in the abdomen.

• Referring the word OLD stallino you are absolutely true and we apoligize for the error that we deleted

in the text.

• we do not have laparoscopic images because at the first laparoscopic examination it was possible to

visualize only the apparently normal part of the testicle. The testicle was then isolated with ligasure

and only after it was completely removed and extracted did we realize that it looked abnormal.

• from a clinical point of view the post-operative did not have any type of complication, but in our

routine we always administer a "protective" dose of antibiotics in all horses undergoing laparoscopy.

We remain at your complete disposal for any further need.

Many thanks

Leonardo Leonardi et al.

Round 2

Reviewer 1 Report

line 33  consists   change to consist

line 38 to 40, I think is repetitive

Author Response

Dear Reviewer,

thank you very much for your kind precious suggestions.

We correct all your remarked parts (line 33  consists   change to consist

line 38 to 40, I think is repetitive) on hope that this versione will be improved.

Unfortunately I cannot read your name, but thanks again and best regards.

Leonardo Leonardi

Reviewer 3 Report

Some of the remarks made last time are still not approach, such as the clinical status of the horse, what happen with the scrotal testicle? was it also analyzed? why did the horse receive 5 days of antibiotics? were there any complications?

The text needs full English proof correction and editing.

Author Response

Dear Reviewer,

thank you very much for your precious support and advices.

We send back the correct form of the paper on the basis also of all your suggestions (

Some of the remarks made last time are still not approach, such as the clinical status of the horse, what happen with the scrotal testicle? was it also analyzed? why did the horse receive 5 days of antibiotics? were there any complications?

The text needs full English proof correction and editing.). We hope this form will work better. 

I cannot read your name, but I thank you again with all my best regards,

Leonardo Leonardi

Round 3

Reviewer 3 Report

Dear authors,

There are still several problems with the language, phrases are not clear, too long, inadequate use of English. Several mistakes keep showing after several suggestions.

The text most be corrected by a professional English editor, and I strongly suggest that you submit a proof of this, as it is very tedious to read again and again the same mistakes. Maybe it is a problem with the PDF software? can you access all comments and suggestions??

The clinical case is very interesting. The hystopathological analysis, I assumed, it has been written by a different person, as it is very clear. But the quality of the manuscript is poor.

Author Response

Dear Reviewer 3, we spent all our effort to try to improve the quality of the paper also involving an English editor. We hope that we have been able to answer to all your kind and true suggestions. Thank you very much again.

Round 4

Reviewer 3 Report

Few minor changes on the following Lines 52, 56, 91, 121, 134-136, 138, 142-143, please see PDFv4 for details

Author Response

Dear Reviewer, we apply all minor changes you suggested in your last review:

Few minor changes on the following Lines 52, 56, 91, 121, 134-136, 138, 142-143, please see PDFv4 for details

Thanks again and best greetings.
